# Lupus, DNA Methylation, and Air Pollution: A Malicious Triad

**DOI:** 10.3390/ijerph192215050

**Published:** 2022-11-15

**Authors:** Leen Rasking, Céline Roelens, Ben Sprangers, Bernard Thienpont, Tim S. Nawrot, Katrien De Vusser

**Affiliations:** 1Centre for Environmental Sciences, Hasselt University, 3590 Diepenbeek, Belgium; 2Depatment of Nephrology and Kidney Transplantation, University Hospital Leuven, 3000 Leuven, Belgium; 3Department of Microbiology and Immunology, Leuven University, 3000 Leuven, Belgium; 4Department of Human Genetics, Leuven University, 3000 Leuven, Belgium; 5Department of Public Health and Primary Care, Environment and Health Unit, Leuven University, 3000 Leuven, Belgium

**Keywords:** systemic lupus erythematosus, lupus, DNA methylation, interferon, air pollution, particulate matter, black carbon

## Abstract

The pathogenesis of systemic lupus erythematosus (SLE) remains elusive to this day; however, genetic, epigenetic, and environmental factors have been implicated to be involved in disease pathogenesis. Recently, it was demonstrated that in systemic lupus erythematosus (SLE) patients, interferon-regulated genes are hypomethylated in naïve CD4^+^ T cells, CD19^+^ B lymphocytes, and CD14^+^ monocytes. This suggests that interferon-regulated genes may have been epigenetically poised in SLE patients for rapid expression upon stimulation by different environmental factors. Additionally, environmental studies have identified DNA (hypo)methylation changes as a potential mechanism of environmentally induced health effects in utero, during childhood and in adults. Finally, epidemiologic studies have firmly established air pollution as a crucial SLE risk factor, as studies showed an association between fine particulate matter (PM_2.5_) and traditional SLE biomarkers related to disease flare, hospital admissions, and an increased SLEDAI score. In this review, the relationship between aberrant epigenetic regulation, the environment, and the development of SLE will be discussed.

## 1. Introduction

Systemic lupus erythematosus (SLE) is a chronic relapsing, a multisystem autoimmune disorder that mainly affects women in their reproductive years. The estimated incidence is 1 to 25 per 100,000 persons in the United States and Europe [1]. SLE is associated with a broad spectrum of clinical manifestations, of which lupus nephritis has the highest morbidity and mortality [2]. SLE is more frequent among non-white populations, with the highest prevalence reported among African-Carribeans [1,3]. Furthermore, it is characterized by the presence of autoreactive T and B lymphocytes and autoantibody production against nuclear and/or cytoplasmic antigens [4]. Pathogenesis of SLE has already been extensively studied [5,6,7,8,9]; however, the precise mechanism remains unresolved until this day [10]. B and T lymphocyte abnormalities [11], dysregulation of apoptosis [12], defects in the clearance of apoptotic material [12], and various (epi)genetic factors [13] have all been implicated in the development of SLE. Here, we review potential culprits in the pathogenesis of SLE, focusing on genetic predisposition, epigenetic alterations, and specific exogenous triggers, such as environmental factors.

## 2. The Role of Genetic Predisposition in the Pathogenesis of SLE

Three lines of evidence suggest that genetic factors may contribute to SLE predisposition, progression, and outcome. First, there are substantial differences in SLE incidence between people of different descent. Epidemiological studies have shown that in men, the incidence is 1.75 times higher in African Americans compared to European Americans; in women, African Americans are 2.6 times more affected [3]. Secondly, clear evidence for familial aggregation was found, with the relative risk (95% confidence interval [CI]) for SLE amounting to 23.68 (20.13–27.84), 11.44 (9.74–13.43), and 14.42 (12.45–16.70) for siblings, for parents, and for offspring of an SLE index patient, respectively [14]. Lastly, twin studies have confirmed genetic susceptibility, as monozygotic twins have a tenfold higher risk of SLE concordance than siblings [15].

A monogenic cause for SLE is established in only one to four percent of SLE patients, including complement system deficiency [16], apoptosis deficiency [16,17], and interferon overproduction [16,18]. In the vast majority, SLE heritability can be explained by a multigenic inheritance model, and genome-wide association studies (GWAS) have revealed over 30 genetic variants associated with SLE, including in loci encoding human leukocyte antigen (HLA), Fc-γ receptor genes, the interferon regulatory factor 5 (*IRF5*) gene, the signal transducer and activator of transcription 4 (*STAT4*) gene, the protein tyrosine phosphatase non-receptor type 22 (*PTPN22*) gene, the tumor necrosis factor alpha-induced protein 3 (*TNFAIP3*) gene, the B lymphocyte kinase (*BLK*) gene, the B lymphocyte scaffold protein with ankyrin repeats 1 (*BANK1*) gene, the tumor necrosis factor superfamily member 4 (*TNFSF4*) gene, and the integrin alpha M (*ITGAM*) gene [19,20,21]. Of these, the strongest association was established in the HLA-encoding region [19,20].

Proteins encoded by SLE-associated genes participate in various pathways and cell types. Specifically, they often affect T lymphocytes (CD4^+^), B lymphocytes (CD19^+^), and monocytes (CD14^+^) (Table 1). They contribute to the etiology of SLE by impacting type I interferons, toll-like receptors (TLR), and the NFκB signaling pathway. Furthermore, these proteins can influence pathways such as apoptosis and the clearance of immune complexes and cellular debris [17,19].

However, the precise biological and functional contribution of most genetic variants remains unestablished to date. The majority of variants associated with SLE susceptibility only cause a moderate increase in SLE risk [15], and only 20% of SLE concordance has been shown in monozygotic twins [14,16]. Hence, genetic susceptibility is unquestionably not the only culprit in SLE pathogenesis. Subsequently, factors other than genetic variation must play a crucial role, such as epigenetic changes or exogenous triggers. Deoxyribonucleic acid (DNA) methylation is one of the most studied epigenetic changes in SLE due to its dependence on, e.g., environmental factors, diet, and/or smoking, as it may link external conditions with gene expression [31].

## 3. The Role of Aberrant DNA Methylation in the Pathogenesis of SLE

### 3.1. DNA Methylation: An Introduction

Epigenetics refers to the array of modifications to DNA and histones which make up our chromatin; it controls gene expression without changing the underlying genome sequence. Epigenetic changes are often triggered by developmental, environmental, or pathogenic stimuli and provide an interface between the environment and gene expression. Subsequently, epigenetic changes can generally be clustered into two main categories: DNA methylation and histone modifications [32,33]. In this review, the focus lies on DNA methylation. However, it is important to note that DNA methylation and histone modifications usually reflect and influence each other.

DNA methylation involves the addition of a methyl group to the 5′ carbon position in the pyrimidine ring of cytosine, generating 5-methylcytosine (5mC). This occurs almost exclusively at cytosine nucleobase located in the context of a CpG dinucleotide, which is relatively rare in our genome, except at specific loci where they are clustered into CpG islands. These CpG islands are typically unmethylated and often colocalize with promoter regions in the genome. CpG island methylation in the promoter region blocks the accessibility of transcriptional activators, thereby inhibiting gene transcription. DNA methylation thus triggers gene silencing and is described as a repressive ‘lock.’ Conversely, an unmethylated state at the promoter region is permissive to gene transcription initiation [34]. The methyl group required for DNA methylation is donated by S-adenosyl-L-methionine (SAM), and the methylation reaction in catalyzed by members of the DNA methyltransferase (DNMT) family (Figure 1).

The DNMT family consists of four catalytically active members in humans: DNMT1, DNMT2, DNMT3A, and DNMT3B [35,36]. DNMT1 predominantly recognizes and copies pre-existing methylation profiles, whereas DNMT3A and DNMT3B induce de novo methylation [35]. The biological functions of DNMT2 remain elusive to this day, but it likely targets RNA, not DNA [36]. Although DNA methylation is relatively stable, the process is also reversible. The demethylation process can occur either in an active or passive manner. Active DNA methylation is independent of DNA replication and requires enzyme activity. Direct demethylation of 5mC to cytosine does not occur, despite earlier claims [37]. Instead, in vivo conversion of 5mC to cytosine involves the replacement of the methylated nucleotide. The first step in this demethylation process is 5mC oxidation by the ten-eleven translocation methylcytosine dioxygenases (TET) to generate 5-hydroxymethylcytosine and further oxidized 5-formyl and 5-carboxycytosine. In the second step, the modified cytosine is replaced with a new, unmodified cytosine through DNA damage repair pathways, presumable involving base excision repair [37]. The passive process occurs during the DNA replication process when DNMT1 is halted, and the copying of the methylation profiles on the new strand fails. The newly synthesized strand is thus not methylated, and the methylation mark is mitigated over time [37,38]. In a third hybrid process, 5mC is first converted to 5hmC by TETs. This base is not recognized by DNMT1 during replication, leading to a failure to copy the DNA methylation mark onto the newly synthesized DNA strand.

### 3.2. The Role of DNA Methylation in SLE

In the last decade, epigenetics has been suggested to play a crucial role in changes in DNA methylation in the onset and development of SLE [39,40]. The first evidence of epigenetic regulation in SLE was found when two drugs, procainamide and hydralazine—both DNA methylation inhibitors—were shown to induce an SLE-like syndrome after prolonged administration in wild-type mice [41]. The same drugs also contributed to disease in a lupus-prone MRLI/lpr mouse model [42]. Furthermore, Sawalha et al. showed that alteration of inhibition of any of the proteins involved in the extracellular signal-regulated kinase (ERK) pathway eventually would lead to the downregulation of DNMTs, inducing anti-double stranded DNA antibody production—resulting in a lupus-like gene expression profile in mice [43].

In humans, a role for DNA methylation in SLE was suggested in a study with monozygotic twins discordant for SLE. The genome-wide DNA methylation pattern in peripheral blood leukocytes showed a lower overall DNA methylation level in the SLE-affected twin [44]. In twins discordant for rheumatoid arthritis (RA) and dermatomyositis, no differences in methylation profile were observed. Notably, peripheral blood leukocytes constitute a mixture of different immune cell types, rending such studies unable to discriminate between cell-type-specific DNA methylation changes and changes in cell type composition. Ulff-Møller et al. [24] showed that in twins discordant for SLE, major cellular compartments, including T and B lymphocytes, monocytes, and granulocytes, displayed marked hypomethylation in the SLE-affected twin.

As epigenetics regulates cell- and tissue-specific gene expression, the disruption of epigenetic events could result in significantly perturbed tissue homeostasis and damage [45]. Furthermore, research has already indicated that some alterations in the epigenome directly contribute to tissue damage and disease expression rather than be a result of chronic inflammation or secondary inflammatory events [45,46]. Moreover, female predominance during the reproductive phase is a key characteristic of SLE; evidence suggests this may be partially due to increased estrogen levels during the reproductive phase [47,48], as Liu and colleagues [49] demonstrated reduced DNA methylation within the estrogen receptor 1 (*ESR1*) gene in women with SLE and RA patients in comparison to healthy controls. This suggests the involvement of reduced DNA methylation, causing increased estrogen signaling in the pathogenesis of SLE.

#### 3.2.1. Interferon Gene Signature

The pathophysiology of SLE is complex and predominantly caused by an inflammatory response to immunogenic endogenous chromatin, acting as a damage-associated molecular pattern [50]. In SLE patients, the chromatin accumulates due to insufficient DNA clearance [51]. This nuclear material activates DNA and RNA sensors in the endosomes and the cytosol of innate immune cells and B lymphocytes; as a result, they produce increased levels of type I interferon (IFN-I) and pro-inflammatory cytokines [52]. Disease activity of SLE and Lupus nephritis can be tracked by monitoring the peripheral blood type I interferon signature [53]. B and T lymphocytes, however, also play an important role in SLE pathogenesis. Immature dendritic cells are activated to become mature, which activates autoreactive T lymphocytes. B lymphocytes persistently produce auto-antibodies (anti-DNA antibodies) as a response to endogenous chromatin-bearing ligands and mediators, such as B-cell activating factor (BAFF) [54]. One hallmark diagnostic feature of SLE includes the presence of these anti-DNA antibodies.

Interferons are a group of signaling proteins, cytokines, which may interfere with and suppress viral replication upon release by a virus-infected cell. Typically, three types of IFNs can be distinguished: type I (e.g., IFN-α and IFN-β), type II IFN (IFN-γ), and type III IFN (IFN-λ). Type I IFNs are the largest family, which can be classified into 5 types: α, β, ε, κ, and ω; IFN-α is the largest subtype, further divided into 12 subtypes [55]. The hallmark immune signature of SLE is type I interferon signaling. Type I IFNs, mainly IFN-α, play a major role in the activation of the innate and adaptive immune system, which are highly dysregulated in SLE. All type I IFN cytokines bind the type I IFN-α receptor, and while IFN-β can be produced by any cell, IFN-α is produced mainly by a specific subpopulation of dendritic cells [56]. Stimulation of IFN-α results in the upregulation of major histocompatibility complex (MHC), as well as costimulatory molecules, which increase survival and activation of dendritic cells, B lymphocytes, and T lymphocytes [57]. IFN-α is known to enhance the antibody response to soluble antigens, stimulate the production of immunoglobulin subtypes, and induce the production of memory B lymphocytes. Furthermore, IFN-α exerts a direct effect on naïve CD4^+^ T lymphocytes to favor their differentiation towards type I helper T lymphocytes, which in turn will secrete IFN-γ [57].

Genome-wide association studies have reported that an increased IFN-α expression in peripheral blood cells of SLE patients could be observed, further termed the IFN signature. This IFN signature, an increased expression of IFN-I-regulated genes, has been previously reported in SLE patients and is used to distinguish IFN type I from IFN type II and IFN type III [56]. Long since, Hooks et al. [58] demonstrated high IFN-α levels in SLE patients; however, pediatric SLE patients displayed an invariable IFN signature at the early stages of the disease, which may suggest the importance of IFN-α in the pathophysiology and disease initiation [53]. One of the hallmarks of SLE includes the formation of immune complexes which activate dendritic cells, increasing antigen-presenting ability, that in turn upregulates IFN-α secretion. On the other hand, IFN-α upregulates dendritic cell maturation as well as upregulation of cell surface molecules; the latter promotes type I helper T lymphocyte response development [59].

Genome-wide DNA methylation analysis performed by Absher et al. [22] demonstrated widespread and severe hypomethylation near genes involved in IFN type I signaling. Furthermore, these IFN-related changes were prominent in active and quiescent stages of the disease; this suggests that the hypersensitivity to IFN mediated through epigenetics persists beyond the acute stages of SLE, acting independently of circulatory IFN levels. The IFN hypersensitivity could be observed in naïve, memory and regulatory T lymphocytes, which suggests that the epigenetics are already established in progenitor cell populations. The IFN signature was also established by Chung et al. [60], who demonstrated that associations with CpG sites within genes either induced by type I IFN or regulated type I IFN signaling could be observed in a large (n = 326) SLE case study.

Ulff-Møller et al. [24] investigated DNA methylation profiles in monozygotic twins with at least one SLE-affected twin. All investigated cell types displayed notable hypomethylation in IFN-regulated genes, such as *IFI44L*, *PARP9*, and *IFITM1*. Furthermore, hypomethylation was more pronounced when SLE-affected twins experienced a disease flare within the past 2 years [24]. Similar results were obtained by Joseph and colleagues [61], who observed significant hypomethylation of differentially methylated sites which were associated with interferon-related genes, including *MX1*, *IFI44L*, *PARP9*, *DT3XL*, *IFIT1*, *IFI44*, *RSAD2*, *PLSCR1*, and *IRF7*. Yeung et al. [62] and Imgenberg-Kreuz et al. [63] also observed hypomethylation in interferon-related genes, including *MX1*, *IFI44L*, and *PLSCR1* [62], and *IRF5*, *IRF7*, *PTPRC*, and *MHC-class III* [63], respectively.

Furthermore, genome-wide DNA methylation analysis by Chen et al. [64] and Coit et al. [23] demonstrated that hypomethylation of IFN-related genes could be considered a common feature of SLE patients in CD4^+^ T lymphocytes, and the DNA methylation profile might be a promising biomarker for diagnostic purposes. Additionally, similar results were obtained by Zhu and colleagues [65], who observed hypomethylation in IFN-regulated genes, revealing significant enrichment in IFN signaling. These findings further establish the importance of IFN in the pathogenesis of SLE.

#### 3.2.2. Specific DNA Methylation in SLE T lymphocytes

In SLE pathogenesis, T lymphocytes have been shown to amplify inflammation through the secretion of proinflammatory cytokines, help B lymphocytes generate autoantibodies, and maintain disease through the accumulation of autoreactive memory T lymphocytes. Furthermore, T lymphocytes involved in SLE pathogenesis can be classified either as pro- or anti-inflammatory. They are known to either drive immunosuppression or inflammation and antibody production, based on the proportion of T lymphocyte subpopulations and their signaling function. In recent years, research focused on the role of different T lymphocyte subsets in the pathogenesis of SLE. Although their prevalence may vary widely, consistent differences in T lymphocyte subpopulation ratios, as well as abnormal functionality of these T lymphocyte subpopulations, are observed in SLE pathogenesis. Altered epigenetic patterns are a hallmark of immune cells involved in SLE pathogenesis. DNA methylation is globally reduced in the T lymphocytes of SLE patients and correlates with disease activity. DNA demethylation plays a central role in the differentiation of T lymphocytes; the interferon-γ locus demethylates upon type 1 CD4^+^ T lymphocyte differentiation, while the IL4/IL5/IL13 locus control region demethylates upon type 2 CD4^+^ T lymphocyte differentiation [66]. In contrast, both loci are heavily methylated in naïve CD4^+^ T lymphocytes [66,67]. Furthermore, DNA hypomethylation can also be observed at the forkhead box P3 (*FOXP3*) locus in regulatory T lymphocytes, compared to naïve T lymphocytes [68].

The CD4^+^ T lymphocytes, also termed helper T lymphocytes (T_H_), aid B lymphocytes in the production of antibodies. CD4^+^ T lymphocytes can be divided into numerous subpopulations, each with their own function within the body, e.g., type 1 (T_H_1) and type 2 (T_H_2) CD4^+^ T lymphocytes [69], T lymphocytes mainly secreting interleukin (IL) 9 (T_H_9) [70], 17 (T_H_17) [71], 22 (T_H_22) [72], follicular T lymphocytes (T_FH_) [73], and regulatory T lymphocytes (T_reg_) [74]. The differentiation of a naïve T lymphocyte into a specific subpopulation is dependent on (i) the antigens that are presented through MHC molecules by antigen-presenting cells (APCs), such as dendritic cells and/or macrophages, and (ii) the cytokine environment provided by APCs. In SLE, CD4^+^ T_H_ lymphocytes contribute to antibody production and tissue inflammation. The most frequently described epigenetic event in CD4^+^ T lymphocytes is global hypomethylation [75]. Numerous cytokine genes have been demonstrated to be epigenetically regulated, leading to hypomethylation in SLE and resulting in the expression of CD4^+^ T lymphocytes. Some of these cytokines have been shown to contribute to tissue damage and/or auto-antibody production, including IL4 [76], IL6 [68], IL10 [77], IL13 [77], and IL17A [45,78].

The relationship between DNA hypomethylation and T cells in autoimmune diseases such as SLE was first elucidated by Richardson et al. [79] Further research discovered that the percentage of 5mC inversely correlated with the disease activity of SLE, potentially illustrating the link between DNA hypomethylation, T cell autoreactivity and SLE [80]. A genome-wide DNA methylation analysis in SLE patients with a history of malar rash, discoid rash or neither revealed 36 and 37 unique DMRs for malar and discoid rash, respectively; the DMRs were primarily localized to cell proliferation and apoptosis-related genes [81]. Furthermore, Lu et al. [82] showed that demethylation of the *CD40LG* gene on the X chromosome in T lymphocytes contributes to its overexpression in women, potentially explaining the higher incidence among women [82].

Qin et al. [83] showed that SLE patients had significantly lower global DNA methylation levels than controls; the global DNA methylation was inversely correlated with the SLE disease activity index [83]. DNA hypomethylation in CD4^+^ T lymphocytes has been shown to activate several genes, including *ITGAL*, *PRF1*, and *TNFSF5/7*, contributing to disease progression and burden [84]. Another study performed in China on 12 healthy donors and 10 SLE patients showed that global DNA methylation in CD4^+^ T lymphocytes in both active and inactive SLE was hypomethylated relative to the control group [85]. Zhao et al. [86] rendered similar results, where specific DNA methylation changes were observed in SLE patients in comparison to age and sex-matched controls. Furthermore, specific DNA methylation changes correlated with the clinical phenotype of SLE (skin lesions with or without chronic renal pathology and polyarticular disease), such as in the genes *NLRP2*, *CD300LB*, and *S1PR3* [86].

Jeffries et al. [87] quantified cytosine methylation at 27,578 CpG sites located within the promoter regions of 14,495 genes in CD4^+^ T lymphocytes. Two hundred thirty-six and 105 of these CpGs were identified as being hypomethylated and hypermethylated, respectively, in SLE CD4^+^ T lymphocytes (n = 12) compared to healthy controls (n = 12). The observed DNA methylation changes were deemed consistent with the widespread DNA methylation changes observed in SLE T lymphocytes. For example, the *CD9* gene was shown to be hypomethylated and is known to provide potent T lymphocyte costimulatory signals, while the transcription factor *RUNX3* was shown to be hypermethylated, which impacts T lymphocyte maturation. Lastly, DNA methylation levels near genes, including *RAB22A*, *STX1B2*, *LGALS3BP*, *DNASE1L1*, and *PREX1,* correlated with disease activity in SLE patients [87].

Coit et al. [23] demonstrated in two independent SLE patient subsets—each consisting of 36 participants—that 86 differentially methylated CpG sites could be identified in 47 genes in comparison to matched healthy control patients. Furthermore, 75% of these CpG sites were hypomethylated. Canonical pathway analysis revealed the interferon signaling pathway as being the most significant. In addition, gene expression analysis indicated significant upregulation of interferon-regulated genes in naïve T lymphocytes, including *IFIT1*, *IFIT3*, *MX1*, *STAT1*, *IFI44L*, *USP18*, *TRIM22*, and *BST2*. Transcription of these genes was increased in the total number of CD4^+^ T lymphocytes in SLE patients. Interestingly, the hypomethylation in these genes was found to be unrelated to disease activity [23]. To further differentiate epigenetic susceptibility loci for lupus nephritis, Coit et al. [28] also investigated genome-wide DNA methylation differences in naïve CD4^+^ T lymphocytes between SLE patients with (n = 28) and without (n = 28) a history of renal involvement and matched healthy controls. One of the most hypomethylated regions included the tyrosine kinase gene *TNK2*, which is involved in cell trafficking and tissue invasion. For all SLE patients, 191 CpG sites in 121 genes could be identified, which were differentially methylated in SLE patients with a history of renal involvement, but not in healthy controls or SLE patients without a history of renal involvement. Furthermore, renal involvement was characterized by more robust demethylation in interferon-regulated genes, independent of disease activity [28].

Absher et al. [22] identified 1033 CpG sites in T lymphocytes with highly significant changes in DNA methylation levels among SLE patients (n = 49) compared to healthy controls (n = 58). Most of the events were hypomethylation near genes involved in IFN type I signaling, both during the active and quiescent stages of the disease. Epigenetically mediated hypersensitivity to IFN persists beyond the acute stage of SLE and is independent of circulating IFN levels. This hypersensitivity was apparent in naïve, memory, and regulatory T lymphocytes, which suggests that the epigenetic state is established in progenitor cell populations [22].

#### 3.2.3. Specific DNA Methylation in B lymphocytes

B lymphocytes exert a direct role in the pathogenesis of SLE through the production of autoantibodies. In SLE, autoreactive B lymphocytes are characterized by their incapability to methylate their DNA, prolonging their survival [88]. Consequently, an increasing number of clinical trials have targeted B lymphocytes as a potential form of therapy without a complete understanding of the mechanism(s) leading to SLE. Similar to T lymphocytes, DNA methylation is also involved in B lymphocyte differentiation [89,90]. In the early stages of B lymphocyte differentiation, demethylation occurs in *PAX5*, which plays an important role in B lymphocyte development progression [91].

In a cohort of African-American females [92], epigenetic relationships in peripheral B lymphocytes were investigated; B lymphocytes were subdivided into subtypes which represent naïve lymphocytes, activated B lymphocytes, and isotype-switched memory B lymphocytes. DNA methylation analysis revealed a distinct molecular SLE disease signature consisting of 6,664 differentially methylated loci stratified in all SLE and control patients. While some CpG sites were hypermethylated in B lymphocyte subsets in sites including *SOX12*, *ARFGAP3*, and *MEG3*, B lymphocyte subsets were predominantly hypomethylated in sites near, IFI44, *IFITM1*, and *YBX1*. In an independent cohort, similar results were obtained where *EPSTI1, IFITM1*, and *MX1* loci were demethylated in SLE patients compared to healthy persons [92]. Absher et al. [22] identified 166 CpG sites in CD19^+^ B lymphocytes in SLE patients (n = 49) compared to healthy controls (n = 58) in a genome-wide DNA methylation analysis.

In twins discordant for SLE, 1628 genes could be identified with differentially methylated CpG sites. B lymphocytes, as all other investigated cell types, displayed a marked hypomethylation in IFN-regulated genes, including *PARP9* and *IFI44L* [24]. One study by Chung et al. [60] examined the association of DNA methylation and SLE-related autoantibodies produced by B lymphocytes. In 326 women with SLE, 467,314 CpG sites were investigated, and the authors identified significant associations between anti-dsDNA autoantibody production and the methylation status of 16 CpG sites in 11 genes, including *IFI44L* and *PARP9/14*. The aforementioned CpG sites were found to be hypomethylated in autoantibody-positive compared to autoantibody-negative cases [60].

B lymphocytes can be referred to as the CD5-nonexpressing B lymphocytes (type II or B2) and the CD5-expressing B lymphocytes (type I or B1). B2 lymphocytes constitute the majority of B lymphocytes, activated to collaborate with helper T lymphocytes to generate specific antibodies. CD5-expressing B lymphocytes are involved in the production of antibodies, including immunoglobulin (Ig) M and binding of a variety of antigens, both self and foreign [93,94]. However, a subset of B1 lymphocytes are shown to be CD5^-^ (B1b) but display the characteristics, functional and phenotypic attributes of a CD5^+^ B lymphocyte (B1a). CD5-nonexpressing B lymphocytes are known to produce autoimmune antibodies, such as SLE [94].

The B lymphocytes from SLE patients are characterized by reduced CD5 expression levels, especially in CD5-nonexpressing B lymphocytes; this promotes their autoreactivity [30]. The CD5-E1B isoform—retained in the cytoplasm—is demethylated in SLE patients’ B lymphocytes in comparison to healthy controls [30]. The same demethylation pattern of CpG islands, which is seen in the CD5 promotor region of SLE B lymphocytes, can be induced in healthy controls by stimulating B lymphocytes with IL6 as well as treatment with the methylation inhibitor PD98059 [30]. Furthermore, when B lymphocytes produce high IL6 levels, the ability of B lymphocytes to induce DNMT1 to methylate DNA is abolished [30]. The importance of IL6 is further corroborated in other studies, including early clinical trials [95,96,97,98], neutralizing cytokines that promote B lymphocyte responses such as IL6 [96].

Among normal B lymphocytes, a reduced capacity to methylate DNA is observed in the autoreactive CD5^+^ B lymphocyte subpopulation, which leads to the expression of repressed genes such as the human endogenous retrovirus (*HERV*) [88]. Fali et al. [95] showed that B lymphocytes could be characterized by their inability to methylate the promoter of the prototype of HERV, HRES-1. In turn, expression of HRES-1 is increased in B lymphocytes after B lymphocyte receptor (BCR) engagement in SLE patients in comparison to healthy controls. Moreover, the authors showed that the Erk/DNMT1 pathway appeared to be defective in SLE B lymphocytes. When the autocrine loop of IL6 is blocked in B lymphocytes, DNA methylation is restored, and HRES-1 expression can be controlled effectively [95].

#### 3.2.4. Specific DNA Methylation in Monocytes and Dendritic Cells

Monocytes are a crucial component of the innate immune system and have been gaining attention in unraveling SLE. Monocytes perform ample functions, including antigen presentation, phagocytosis, and cytokine production, resulting in the recognition of SLE disease development and progression [99]. In SLE patients, monocytes share several hypomethylated CpG sites with CD4^+^ T lymphocytes, although they are more prominently and numerously hypomethylated in T lymphocytes. For example, Absher and colleagues [22] identified 97 CpG sites that had methylation changes in monocytes in SLE patients compared to healthy controls and showed 27 genes in monocytes to be strongly associated with SLE, while Ulf-Møller and colleagues [24] found 327 differentially methylated CpG sites in monocytes and 247 in granulocytes. However, to date, research investigating DNA methylation changes in monocytes is still scarce.

Dendritic cells (DCs) are antigen-presenting cells that act as a messenger between the innate and adaptive immune systems. Under normal conditions, DCs clear apoptotic debris, recognize self-DNA and RNA through Toll-like receptors (TLR) and promote B lymphocyte maturation and proliferation through secretion of IFN-α. Their uncontrolled activation might drive autoimmune diseases such as SLE; moreover, DCs have been implicated in SLE pathogenesis [100,101]. However, to date, little research focused on DC DNA methylation in SLE.

## 4. The Role of Air Pollution in the Pathogenesis of SLE

### 4.1. Air Pollution: An Introduction

Air pollution is a significant environmental risk factor for human health, estimated to cause approximately 800,000 premature deaths worldwide annually [102]. According to the World Health Organization (WHO), in 2019, 99% of the world’s population lived in areas where the air quality guidelines were not met [103]. Moreover, air pollution is already extensively linked to, e.g., respiratory disease [104,105], cardiovascular disease [106,107], and cancer [108] (Figure 2). Ambient air pollution majorly consists of gases such as ozone (O_3_), sulfur dioxide (SO_2_), nitrogen dioxide (NO_2_), and carbon monoxide (CO), along with particulate matter (PM) [109]. PM is a complex mixture consisting of liquid and solid particles suspended in the air [110]. It exists in various sizes, chemical compositions, surface areas, and masses [110,111], which play a crucial role in the consequent health impacts of PM [112]. Moreover, the aerodynamic diameter is most often employed to categorize the particles, ranging from coarse (PM_10_; diameter ≤ 10 microns) to fine (PM_2.5;_ diameter ≤ 2.5 microns) and ultrafine (PM_1.0_; diameter ≤ 1 micron) particles.

Furthermore, animal and in vitro studies have shown that small particulates, such as diesel exhaust particles (DEP), may penetrate the lung barrier, alter the cell’s function, and enter blood circulation [113,114,115]. Research has indicated that PM can cause effects similar to those observed after exposure to inhaled cigarette smoke [116] or silica [117] on the immune system [118,119], inducing inflammation and oxidative stress [120,121]. The most prominent toxic component of PM is the combustion-derived particles, or so-called soot, which are generated during the incomplete combustion of fuels and include, among others, the environmental contaminant black carbon (BC) [103]. Moreover, SO_2_ is deemed interdependent with particulate matter, as it is considered primarily derived from coal combustion [122].

Ozone is considered a photochemical oxidant, formed as a secondary pollutant through solar radiation in the presence of primary pollutants (e.g., nitrogen oxides or volatile organic compounds) [123]. In the presence of its precursor primary pollutant, such as NO, O_3_ is scavenged, leading to lower concentrations, where high NO concentrations are measured and vice versa [124]. NO_2_ is converted from NO through oxidation reactions that involve oxygen and O_3_ [122,124]; it is mainly produced as a result of vehicle emission and, therefore, a well-known indicator for ambient traffic-generated air pollution [122,125]. Lastly, CO is mainly produced through the incomplete combustion of carbonaceous fuels and, therefore, mainly emitted from vehicles. Concentrations have been found to be relatively high in traffic-dense areas, including road tunnels or car parks [122,123].

Even though the health effects following ambient air pollution are well-described, the understanding of the biological mechanism(s) mediating these health effects remains elusive to date. Recently, epigenetics has gained attention as potentially driving these exposure-disease associations. 

**Figure 2 ijerph-19-15050-f002:**
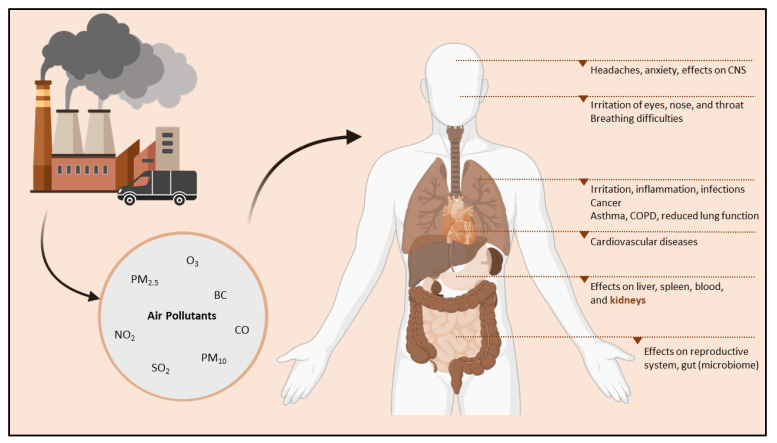
Air pollutants derived from the air pollution generated by, e.g., industry and vehicle emissions negatively influence organs beyond the lungs, possibly causing, e.g., cardiovascular diseases and exerting effects on the central nervous system, liver, kidneys, spleen, and blood. Image created with Biorender^®^. Abbreviations: BC, black carbon; CNS, central nervous system; CO, carbon monoxide; COPD, chronic obstructive pulmonary disease; NO_2_, nitrogen dioxide; O_3_, ozone; PM_2.5_, fine particulate matter; PM_10_, coarse particulate matter; SO_2_, sulfur dioxide.

### 4.2. Air Pollution and DNA Methylation

The cause-effect relationship for genes to either undergo hypo- or hypermethylation depends on the type of environmental cue received [126]. PM comprises heavy metals and polycyclic aromatic hydrocarbons (PAHs); they are responsible for a feed-forward loop in the oxygen-radical-mediated pro-inflammatory cytokine response. Deregulation of several transcription factors might play an important role in the methylation cascade [127]. Furthermore, PM-induced reactive oxygen species (ROS) oxidize 5-hydroxy-mC, causing DNA methylation; moreover, in the presence of ROS molecules, histone deacetylase (SIRT1) interacts with DNMT1, inducing variations in DNA methylation levels [128]. PM also prohibits the ability of DNMTs to function normally by aligning with the DNA, leading to CpG cytosine residue hypomethylation. This, in turn, leads to altered DNA methylation machinery [129]. Additionally, any metal components present in PM may increase its potential to induce deregulatory effects on methylation levels, such as environmental arsenic has been linked to hypomethylation levels in blood DNA by Tellez-Plaza et al. [130] Similar effects were observed when investigating lead [131] and cadmium [132].

#### 4.2.1. The Effects of Air Pollution on DNA Methylation In-Utero and during Childhood

Emerging evidence has suggested DNA methylation changes as a potential pathway of environmentally induced health effects. DNA methylation changes have been extensively studied regarding air pollution exposure across the life course, from in-utero to adulthood. The ENVIR*ON*AGE birth cohort is an ongoing population-based prospective birth cohort study that studies the health effects of air pollution exposure during the most vulnerable stages of life, from birth until preadolescent age. Mother-newborn pairs are recruited when they arrive for delivery at the East-Limburg Hospital in Genk (Belgium). Currently, over two-thousand mother-newborn pairs have been recruited from 2010 onward, making this the largest birth cohort with a prospective follow-up in Belgium [133]. In this cohort, Janssen et al. observed a lower degree of global placental DNA methylation associated with PM_2.5_ exposure in early pregnancy [110] and a positive association between an interquartile range increment of PM_2.5_ and mitochondrial DNA methylation in two regions, i.e., the D-loop control region and *MT-RNR1* [134]. Vos et al. [135] quantified DNA methylation levels in two regions of the displacement loop control region (D-loop and low-density lipoprotein receptor 2 [*LDLR2*]), which showed higher D-loop methylation levels for smokers and participants highly exposed to air pollutants. Furthermore, the methylation levels correlated to placental mitochondrial DNA content [135].

Within the same cohort study, Neven et al. [136] showed that transplacental in-utero PM and BC exposure could be associated with epigenetic alterations in key DNA repair and tumor suppressor genes. Additionally, Saenen et al. [137] showed that the methylation status of placental leptin, an energy-regulating hormone involved in fetal growth and development, was negatively associated with PM_2.5_ exposure during the second semester. Furthermore, Nawrot et al. [138] demonstrated that CpG sites within the promoter regions of various genes involved in the circadian pathway, including clock circadian regulator (*CLOCK*)*,* neuronal PAS domain protein 2 (*NPAS2*), period circadian regulator 1–3 (*PER1-3*), and basic helix-loop-helix ARNT like 1 (*BMAL1*), were methylated in the placenta of 407 newborns. For example, placental circadian pathway methylation was positively and significantly associated with PM_2.5_ exposure during the third trimester in placental *NPAS2* and *PER3* per interquartile increase of PM_2.5_ [138].

Similar findings were reported by Maghbooli and colleagues [139], who showed significantly positive correlations between PM_2.5_ exposure in the first trimester of pregnancy and placental global DNA methylation levels [139]. Moreover, prenatal PM_2.5_ exposure might also lead to aberrant DNA methylation changes in the placental genome, mainly observed in reproductive development, energy metabolism, and the immune response. Furthermore, DNA methylation of insulin-like growth factor 1 (*IGF1*) and BH3 interacting domain death agonist (*BID*) showed significant associations with PM_2.5_ exposure [140]. Ladd-Acosta et al. [141] observed global, locus, and sex-specific methylation changes associated with prenatal NO_2_ and O_3_ exposure. The authors identified DNA methylation in six differentially methylated regions (DMRs), of which three are sex-specific in association with prenatal NO_2_ exposure; an additional three sex-specific DMRs could be identified and associated with prenatal O_3_ exposure. DMRs initially detected in cord blood showed consistent exposure-related changed in DNA methylation in the placenta; however, those initially detected in the placenta showed no DNA methylation differences in cord blood—appearing to be tissue-specific [141]. In a Korean birth cohort [142], DNA methylation in association with PM_10_ and NO_2_ exposure were investigated; most significantly, a CpG site in the caspase 7 (*CASP7*) gene could be associated both when addressing the entire pregnancy and during the second trimester. Furthermore, CpG sites related to WD repeat domain 93 (*WDR93*) and family with sequence similarity 176 member A (*FAM176A*) could be associated with PM_10_ and NO_2_ exposure [142].

Kingsley et al. [143] indicated that living close to major roadways could be significantly associated with placental epigenetic changes in relation to fetal growth. A study by Breton and colleagues highlights that the effects of early life exposure and differences in either the type of pollutant or developmental stage at exposure are important key factors. Hereby, NO_2_ exposure during the third trimester of pregnancy was associated with higher systolic blood pressure in eleven-year-old children, but no association could be demonstrated with DNA methylation in the blood’s long interspersed nuclear elements 1 (*LINE1*) gene; however, O_3_ could be associated conversely with higher DNA methylation of *LINE1*. PM_10_ and O_3_ exposure during the first trimester were associated with lower DNA methylation of *LINE1* at birth; the latter exposure could also be associated with increased systolic blood pressure in 11-year-old children with specific DNMT1 or DNMT3B isoforms [144]. In addition, Cai et al. [145] stipulated that PM_10_ exposure could also be associated with DNA methylation of *LINE1* and hydroxysteroid 11-β dehydrogenase 2 (*HSD11B2*) during early pregnancy, which may mediate PM-induced reproductive and developmental toxicity [145].

Gruzieva et al. showed that DNA methylation in newborns of three CpG sites in mitochondria-related genes could be associated with NO_2_ exposure during pregnancy [146]. The aforementioned study also investigated DNA methylation in (older) children, with DNA methylation of one of the CpG sites in mitochondria-related genes remaining significant in older children in association with NO_2_ exposure [146]. However, it is of note that the NO_2_ exposure at the time of biosampling is deemed important. Another large meta-analysis by Gruzieva et al. [147] showed that several differentially methylated CpG sites and differentially methylated regions (DMRs) could be associated with prenatal PM exposure in new-borns. For prenatal PM_10_ and PM_2.5_ exposure, six and 14 CpG sites were significantly associated, respectively. Two PM_10_-related DMRs investigated in older children, i.e., H19 imprinted maternally expressed transcript (*H19*) and membrane-associated ring-CH-type finger 1 (*MARCH11*), also replicated in new-borns [147]. Isaevska et al. [148] investigated DNA methylation in cord blood in relation to gestational PM_10_ exposure, which could be associated with the DNA methylation of >250 unique differentially methylated probes (DMP). Most of these DMPs could be identified in early gestation; eight showed robust associations with PM_10_ exposure during early gestation, while two with PM_10_ exposure during the whole pregnancy [148]. 

Abraham and colleagues [149] showed that air pollutants, including PM_10_ and NO_2_, could be associated with 27 DMRs, of which some are involved with genes that are implicated in pre-eclampsia, hypertensive and metabolic disorders, such as adenosine A2b receptor (*ADORA2B*). Nine CpG sites were identified, which mapped to nine genes associated with prenatal exposure to PM_10_ and NO_2_; The methylation of two CpG sites located in *ADORA2B* remained significantly associated during the whole pregnancy [149]. Additionally, placental DNA methylation was shown to mediate early-onset atopic dermatitis in association with higher PM_2.5_ exposure during the first trimester of pregnancy, paired with low cord blood vitamin D levels [150].

The abovementioned findings suggest that air pollution exposure-induced changes during pregnancy may persist well into childhood; furthermore, a potential mechanism for DNA methylation modulation following air pollution exposure could involve changes in the expression of key enzymes which regulate DNA methylation [151].

#### 4.2.2. The Effects of Air Pollution on DNA Methylation in Adults

The effects air pollution exerts on DNA methylation have also been investigated in adults. In a study of older Bostonian male volunteers assessing the effects of PM, BC, and O_3_ exposure on blood DNA methylation, lower DNA methylation was observed, associated with five immune-related genes, such as IL6, coagulation factor III tissue factor (F3), IFN-γ, and intercellular adhesion molecule (ICAM) 1 [152]. An 18% decrease in DNA methylation of the *F3* gene was associated with an interquartile increase of particles (±1.5 × 10^4^ particles per mm³), albeit without cell-type correction. Increased O_3_ exposure two to four weeks prior to the clinic visit could be associated with lower DNA methylation in the promoter region of *ICAM1* [152]. In a follow-up analysis, the authors separated the methylation values into quantiles according to the volunteer’s degree of pre-existing DNA methylation and correlation with air pollution for these quantiles rather than a mean value [153]. Stronger negative associations in DNA methylation for the *F3* gene and particle number could be observed; furthermore, a positive association was shown between BC exposure and DNA methylation of the *ICAM1* gene at the ninth quantile, but negative associations in the first to sixth quantiles [153].

In whole blood, De Prins and colleagues [154] showed that decreased global DNA methylation could be associated with exposure to NO_2_, PM_10_, PM_2.5_, and O_3_ at the home addresses of non-smoking adults. Furthermore, results demonstrated that men had higher DNA methylation in both summer and winter in comparison to women. When analyzing the data for winter and summer together, various moving average exposures for NO_2_, PM_10_, and PM_2.5_ were associated with changes in the percentage of 5-methyl-2′-deoxycytidine (% 5mdC; 95% CI), ranging from −0.04 (−0.09–0.00) to −0.14 (−0.28–0.00) per interquartile range (IQR) increase in pollutant [154]. Mostafavi et al. [155] showed that personal—repeated 24-h personal exposure measurements—PM_2.5_ exposure could be associated with DNA methylation changes at 13 CpG sites and 69 DMRs, with two of the identified CpG sites located in the identified DMRs. Additionally, personal exposure to PM_2.5_ absorbance, ultrafine particles (UFPs), ambient PM_2.5_, ambient PM_2.5_ absorbance, and ambient UFPs were associated with 42, 16, four, 16, and 15 DMRs, respectively [155].

Jiang et al. [156] demonstrated that short-term exposure to diesel exhaust resulted in changes in DNA methylation at CpG sites found in genes that are involved in the inflammation and oxidative stress response. 2827 CpG sites could be identified in which DNA methylation occurred in persons exposed to diesel exhaust but not filtered air; exposure-related DNA methylation changes were observed at sites for genes involved in protein kinase and NFκB pathways [156].

Results demonstrate that air pollution exposure could affect DNA methylation at specific immune system-related CpG sites, which associates with the modulation of associated gene expression.

### 4.3. Air Pollution and SLE Development

It is generally believed that DNA methylation or demethylation does not occur spontaneously, i.e., there must be a trigger or external factor that leads to DNA hypomethylation. Many environmental factors have previously been implicated as SLE triggers, and it is becoming more apparent that the mechanism(s) by which these environmental factors exert their effects is through epigenetics Table 2.

For example, in childhood-onset SLE patients [157], an increased risk of an elevated SLE disease activity index 2000 (SLEDAI-2K) score (≥8) could be associated with the PM_2.5_ 7-day moving average [157]. In another childhood-onset SLE panel study [158], each IQR increase (18.12 µg/m³) in PM_2.5_ exposure was associated with an increased risk of nephritis and positive anti-dsDNA results. Furthermore, a decrease in C3 serum levels and an increase in 24 h-urinary protein could also be associated with NO_2_ and PM_2.5_ exposure, respectively [158]. Conde et al. [159] also determined that environmental factors such as PM_10_, SO_2_, NO_2_, O_3_, and CO were risk factors for the development of SLE during childhood. Furthermore, in a larger population study over a longer time period, the incidence rate for childhood-onset SLE increased almost fivefold when comparing the first quartile to the fourth quartile of PM_2.5_ exposure [160]. Campos et al. [161] showed the effects of air pollutants on disease activity in childhood-onset SLE; IQR increases of PM_10_, CO, and NO_2_ could be associated with an increased risk of a SLEDAI-2K score higher than 8, while O_3_ and SO_2_ seemed to have no apparent effect [161].

The effects of air pollutants on the development and/or exacerbation of clinical aspects of SLE alongside hospital admissions attributable to SLE have also been investigated in adults. In a Canadian cohort assessing two provinces [162], PM_2.5_ exposure could be associated with the development of systemic autoimmune rheumatic diseases (SARDs), with a higher probability among older females [162]. A Taiwanese study by Jung et al. [163] showed that positive associations for SLE development could be observed with an increase in NO_2_, CO, and PM_2.5_, while O_3_ and SO_2_ were negatively associated [163]. In Chile, Cakmak et al. [164] demonstrated that SO_2_, CO, and PM_2.5_ were positively associated with increased hospital admissions with a primary diagnosis of SLE. Similarly, Zhao et al. [165] associated hospital admissions due to SLE with high exposure levels of PM_2.5_, NO_2_, and SO_2_. Furthermore, the Chinese study also indicated that high PM_2.5_ concentrations could be associated with an increased risk of SLE relapse, while high NO_2_ and SO_2_ exposure levels could be associated with an increased risk of being first-time admission to the hospital for SLE [165]. Bernatsky et al. [166] investigated whether PM_2.5_ exposure affected the clinical aspects of SLE, described in the SLEDAI-2K. Two traditional biomarkers for disease flare(s), i.e., urinary casts and anti-dsDNA, were associated with short-term variations in PM_2.5_ exposure shortly before the clinical visits (24 to 48 h before) [166].

One study investigated distance to a major road (motor vehicle emission exposure), SLE, and DNA methylation. Lanata et al. [167] demonstrated that three methylation sites were significantly hypomethylated in SLE patients residing close to a major highway, of which all three sites belonged to a single gene, *UBE2U*. However, these results could not be replicated in a control cohort, which the authors suggest can be explained by the increased susceptibility of SLE patients [167].

Air pollution and (the development of) SLE have just yet begun to be acknowledged; however, evidence is rising about the detrimental effects air pollutants exert on the development and exacerbation of the disease.

**Table 2 ijerph-19-15050-t002:** Summary of Articles addressing the Effects of Air Pollution on SLE.

Author	SLE Outcome Investigated	Environmental Factor Investigated	Main Findings
Alves et al. [157]	SLEDAI-2K score	7-day moving average of PM_2.5_ exposure	A significant increase in the risk of a SLEDAI-2K score ≥8
Goulart et al. [158]	Nephritis Anti-dsDNA Serum markers, i.e., C3 Urinary markers	PM_2.5_, NO_2_ exposure	PM_2.5_ exposure caused an increased risk of nephritis and rendered positive anti-dsDNA results PM_2.5_ and NO_2_ exposure caused a decrease in serum C3 and an increase in 24 h urinary protein
Conde et al. [159]	SLE development	PM_10_, SO_2_, NO_2_, O_3_, and CO exposure	All investigated environmental factors increased the risk of developing SLE during childhood
Mai et al. [160]	Incidence of childhood-onset SLE	PM_2.5_ exposure	Incidence of childhood-onset SLE increased almost 5-fold comparing the fourth quartile to the first quartile of PM_2.5_ exposure
Campos et al. [161]	Disease activity in childhood-onset SLE	PM_10_, SO_2_, NO_2_, O_3_, and CO exposure	PM_10_, CO, and NO_2_ exposure increases were associated with an increased risk of ≥8 SLEDAI-2K score, O_3_ and SO_2_ exposure had not apparent effect
Bernatsky et al. [162]	Systematic autoimmune rheumatic diseases (SARDs), including SLE	PM_2.5_ exposure	PM_2.5_ exposure caused a higher risk of developing SARDs, with a higher probability among females
Jung et al. [163]	SLE development	PM_10_, SO_2_, NO_2_, O_3_, and CO exposure	Positive associations were found for SLE development with increased NO_2_, CO, and PM_2.5_ exposure, while negative associations were demonstrated for O_3_ and SO_2_ exposure
Cakmak et al. [164]	Hospital admissions with SLE as the primary diagnosis	PM_10_, SO_2_, and CO exposure	All investigated environmental factors were associated with increased hospital admissions with a primary diagnosis of SLE
Zhao et al. [165]	Hospital admissions due to SLE Relapses of SLE	PM_10_, SO_2_, and NO_2_ exposure	More hospital admissions for SLE were observed with increased exposure to PM_2.5_, NO_2_, and SO_2_. Increased PM_2.5_ exposure caused an increased risk of SLE relapse High NO_2_ and SO_2_ exposure levels were associated with an increased risk of first-time admission for SLE
Bernatsky et al. [166]	SLEDAI-2K score Biomarkers for disease flare, i.e., urinary casts anti-dsDNA	PM_2.5_ exposure	Urinary casts and anti-dsDNA were associated with short-term PM_2.5_ variations shortly before the clinical visit(s)
Lanata et al. [167]	Hypomethylation in SLE	PM_2.5_ exposure	Three methylation sites were significantly hypomethylated in SLE patients residing close to a major highway, of which all three sites belonged to a single gene, *UBE2U*.

Abbreviations: CO, carbon monoxide; NO_2_, nitrogen dioxide; O_3_, ozone; PM, particulate matter; SARDs, systemic autoimmune rheumatic diseases; SLE, systemic lupus erythematosus; SLEDAI, SLE disease activity index; SO_2_, sulfur dioxide.

## 5. Conclusions

In this review, we discussed the relationship between aberrant epigenetic regulation and the influence of the environment on the development of SLE. Ample evidence suggests that DNA hypomethylation plays a significant role in SLE development, which is mainly studied in naïve CD4^+^ T lymphocytes but has also been described in (CD19^+^) B lymphocytes and CD14^+^ monocytes. Most of the reported hypomethylated regions are localized in upstream promotor regions of interferon-regulated genes. This suggests that interferon-regulated genes in SLE naïve CD4^+^ T lymphocytes may be epigenetically poised for rapid induction upon stimulation by various factors, including air pollution.

DNA hypomethylation has been suggested as a potential pathway of environmentally induced health effects; convincing evidence suggests that PM_10_ and PM_2.5_ may play a significant role in SLE development. Furthermore, PM might provoke SLE development, stronger immunogenic reactions, and more SLE flares through the induction of DNA methylation changes. Plausibly, environmental factors may be able to prevent the replication of DNA methylation patterns during mitosis, resulting in DNA hypomethylation in T lymphocytes, B lymphocytes, and monocytes. Another probable mechanism could be active DNA demethylation in T lymphocytes, B lymphocytes, and monocytes by environmental factors, such as air pollution. Research suggests IFN-regulated genes in SLE, naïve CD4^+^ T lymphocytes, CD19^+^ B lymphocytes, and monocytes may have resided in an epigenetically poised status before expression, which after stimulation results in rapid expression, resulting in SLE flares [22,23,168].

In a complex disease such as SLE, the identification of novel biomarkers that can predict disease manifestations, activity, flares, and response to therapy is crucial. Outlining DNA hypomethylation in SLE patients versus healthy controls could potentially identify epigenetically modified genetic targets, which could aid in better understanding SLE disease pathogenesis and potentially identify novel targets for therapy. Additionally, the dynamic nature of DNA methylation changes renders them appealing targets to explore as disease activity biomarkers.

It is of note that pollutants in the air may also exert opposite effects, such as the reactivation of the antioxidant system. For example, O_3_ has been shown to exert an oxidative action on plasma proteins in humans when doses were applied in a therapeutical range [169], and its therapy involves blood cells and the endothelium of blood vessels [170]. Furthermore, O_3_ has already shown a protective effect in a rodent model with induced renal ischemia; renal plasma flow and GFR were significantly decreased after ischemia induction and subsequent reperfusion [171].

However, air pollution as a potentially significant risk factor requires further evaluation. The novel white light technique to quantify black carbon particles—an internal exposure marker for air pollution—in biological tissue and biofluid samples as a marker for chronic (>1 month) combustion-derived particle accumulation opens a new window to predict possible disease evolution and the onset or number of flares [172]. Furthermore, Nawrot et al. [173] instigated that homocysteine may be a causal intermediate in the association between the effects of air pollution and epigenetic alterations; therefore, supplementation with folic acid may reduce air pollution-induced methylation.

## Figures and Tables

**Figure 1 ijerph-19-15050-f001:**
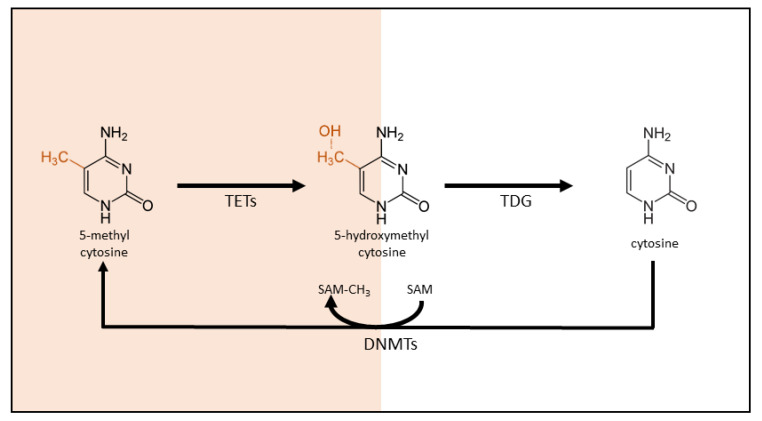
The DNA (de)methylation process. In DNA methylation, cytosine is converted to 5-methylcytosine by DNA methyltransferase (DNMT). Members of the DNMT family catalyze the transfer of a methyl group (CH_3_) from S-adenosyl-L-methionine (SAM) to the 5-carbon position of cytosine. The methyl group required for DNA methylation is donated by SAM. DNA demethylation is a multistep oxidation process catalyzed by the ten-eleven translocation (TET) methylcytosine dioxygenase family.

**Table 1 ijerph-19-15050-t001:** Overview of genes associated with SLE in T lymphocytes, B lymphocytes, monocytes, and neutrophils. Listed genes only represent a fraction of all genes with altered methylation proportions found in SLE [14,22,23,24,25,26,27,28,29,30].

T lymphocytes (CD4^+^)	B lymphocytes (CD19^+^)	Monocytes and Neutrophils (CD14^+^)
PTPN22 (protein tyrosine phosphatase, non-receptor type 22)	BLK (B-lymphocyte kinase)	HLA-DR (human leukocyte antigen-DR isotype)
TNFSF4 (TNF Superfamily Member 4)	BANK1 (B cell scaffold protein with ankyrin repeats 1)	TNFα (tumor necrosis factor α)
STAT4 (Signal transducer and activator of transcription 4)	LYN (LYN Proto-Oncogene, Src Family Tyrosine Kinase)	ICAM1 (intercellular adhesion molecule 1)
CD247 (cluster of differentiation 247)	CR2 (complement receptor 2)	Fc-γ RII (Fc gamma receptors class II)
CD9 (cluster of differentiation 9)	NCF2 (neutrophil cytosol factor 2)	ITGAM (CD11b) (integrin subunit alfa M [cluster of differentiation 11b])
MMP9 (matrix metallopeptidase 9)	IL1 (interleukin 1)	NETs (norepinephrine transporters)
PDGFRA (platelet-derived growth factor receptor A)	IKZF1 (IKAROS family zinc finger 1)	IFI44L (interferon-induced protein 44-like)
Perforin	TLR9 (toll-like receptor 9)	ADAR (adenosine deaminase RNA specific)
CD11a (cluster of differentiation 11a)	CD19 (cluster of differentiation 19)	RABGAP1L (RAB GTPase activating protein 1 like)
CD70 (cluster of differentiation 70)	ISGs (interferon-stimulated genes)	CMPK2 (cytidine/uridine monophosphate kinase 2)
CD40 ligand (cluster of differentiation 40 ligand)	CD5 (cluster of differentiation 5)	TREX1 (three prime repair exonuclease 1)
PP22A (protein phosphatase 2A)	HRES1 (HTVL-1-related endogenous sequence)	DTX3L (deltex E3 ubiquitin ligase 3L)
BST2 (bone marrow stromal cell antigen 2)	TNF (tumor necrosis factor)	PARP9 (poly ADP-ribose polymerase family member 9)
IR7 (ionotropic receptor 7)	EP300 (E1A binding protein P300)	PLSCR1 (phospholipid scramblase 1)
CD80 (cluster of differentiation 80)	IFI44L (interferon-induced protein 44-like)	DDX60 (DExD/H-Box helicase 60)
HERC5 (HECT and RLD domain containing E3 ubiquitin-protein ligase 5)	PARP9 (poly ADP ribose polymerase family member 9)	HLA-F (major histocompatibility complex, class 1, F)
IFI44 (interferon-induced protein 44)	IFITM1 (interferon-induced transmembrane protein 1)	TAP1 (transporter 1, ATP binding cassette subfamily B member)
ISG15/20 (interferon-stimulated gene 15/20)	ISG15 (interferon-stimulated gene 15)	PSMB9 (proteasome 20S subunit beta 9)
ITGAX (integrin subunit alpha X)	PRDM16 (PR domain containing 16)	PARP12 (poly ADP ribose polymerase family member 12)
PARP12 (poly ADP ribose polymerase family member 12)	RCAN3 (RCAN family member 3)	PDE7A (phosphodiesterase 7A)
TNK2 (tyrosine kinase non-receptor 2)	RUNX3 (runt-related transcription factor 3)	IFIT3 (interferon-induced protein with tetratricopeptide repeats 3)
DUSP5 (dual specificity protein phosphatase 5)	FAM167B (family with sequence similarity 167-member B)	IFIT1 (interferon-induced protein with tetratricopeptide repeats 1)
TET3 (TET methylcytosine dioxygenase 3)	IFI44L (interferon-induced protein 44-like)	IFITM1 (interferon-induced transmembrane protein 1)
INPP4A (inositol polyphosphate 4 phosphatase type I A)	PRDX6 (peroxiredoxin 6)	OAS1 (2′-5′-Oligoadenylate synthetase 1)
IL1RN (interleukin 1 receptor antagonist)	RABGAP1L (RAB GTPase activating protein 1 like)	NLRC5 (NLR Family CARD domain containing 5)
ACVR1 (activin A receptor type 1)	CMPK2 (cytidine/uridine monophosphate kinase 2)	RNF213 (ring finger protein 213)
EPHA4 (ephrin type A receptor 4)	RSAD2 (radical S-adenosyl methionine domain containing 2)	ZCCHC2 (zinc finger CCHC-type containing 2)
CCDC12 (Coiled-coil domain-containing protein 12)	EIF2AK2 (eukaryotic translation initiation factor 2 alfa kinase 2)	PRIC285 (peroxisomal proliferator-activated receptor A-interacting complex 285 kDa protein isoform 2)
TREX1 (Three Prime repair exonuclease 1)	VPS54 (Vacuolar protein sorting associated protein 54)	MX1 (MX dynamin-like GTPase 1)
UBA7 (Ubiquitin-like modifier activating enzyme 7)	REG1B (Regenerating family member 1 beta)	GGT1 (γ-glutamyltransferase 1)

## Data Availability

Data sharing is not applicable to this article as no datasets were generated or analyzed during this narrative review.

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
