# Peer review of "Lupus, DNA Methylation, and Air Pollution: A Malicious Triad"

_ijerph, 2022, doi:10.3390/ijerph192215050_

Round 1

Reviewer 1 Report

This review by Rasking and colleagues is interesting, well organized and well written, debating the role of air pollution in SLE susceptibility, which seems to have an increasing key role in causing epigenetic changes. Investigating DNA methylation, a wide number of genes is reported as potentially involved in SLE onset and progression, thus resulting in a detailed description of the state of the art.

Only some comments.

Line 358: I would suggest to delete Ulf-Moller et al., since the reference [24] is reported.

Line 368: Since the Authors described T lymphocytes subpopulations and their role, I would suggest to (briefly) describe also the CD5 subpopulation in B lymphocytes.

Line 425: The Authors illustrated the role of DEP as one of the main culprit of air pollution. In this regard, I would suggest to report also the recent work by Colasanti et al. "Diesel exhaust particles induce autophagy and citrullination in Normal Human Bronchial Epithelial cells" (Cell Death Dis. 2018 Oct 19;9(11):1073. doi: 10.1038/s41419-018-1111-y), in which an alteration of bronchial epithelial cell function is showed after DEP exposure. Consequently, the sentence (lines 425-426) could change as follows: "Furthermore, animal and in vitro studies have shown that small particulates, such as e.g., diesel exhaust particles (DEP) may penetrate the lung barrier, alter the cell function and enter the blood circulation".

Figure 2 legend, line 449: I would replace "including" with "possibly causing".

In the paragraph 4.2.1. "The Effects of Air Pollution on DNA Methylation in-utero and during childhood" genes are listed as acronyms, but their complete name is not reported, rendering difficult to understand their involvement.

In Table 2, I would suggest to list the environmental factors investigated as "PM2.5, NO2, SO2, O3 exposure (and so on)", without repeating "exposure" for each. The Table will result less crowded.

Please check all the abbreviations (they have to be written out in full only on first use).

Author Response

Dear Editor, Dear Reviewer,

Please see the attachment for the point-by-point response to the raised comments and concerns.

Thank you in advance,

Leen Rasking and Katrien De Vusser on behalf of all authors

Reviewer 2 Report

It is a very well-written review text, easy to understand, and didactic to the reader. Language is clear and scientific. The references are well-used, well updated, and adequate. However, the text does not have articles that oppose it, that is, it only has papers showing the authors' point of view on the issue that can induce the reader to accept all the arguments. An example is the effect of ozone. See below some articles, e.g., that contradict the statements of the text. So, I recommend that the authors review and add some points against their statements, as the entire text is aimed only at one line of thought without in view of opposing opinions. This would be the fairest and most scientific way to approach this review. Congratulations!

Some examples:

Bocci V. Is it true that ozone is always toxic? The end of a dogma. Toxicol Appl Pharmacol. 2006 Nov 1;216(3):493-504. doi: 10.1016/j.taap.2006.06.009. Epub 2006 Jun 27. PMID: 16890971.

Bocci V, Borrelli E, Travagli V, Zanardi I. The ozone paradox: ozone is a strong oxidant as well as a medical drug. Med Res Rev. 2009 Jul;29(4):646-82. doi: 10.1002/med.20150. PMID: 19260079.

Author Response

(The authors gave the same response as above.)
